# Prognostic Value of Plasma Biomarkers S100B and Osteopontin in Pediatric TBI: A Prospective Analysis Evaluating Acute and 6-Month Outcomes after Mild to Severe TBI

**DOI:** 10.3390/biomedicines11082167

**Published:** 2023-08-01

**Authors:** Laura S. Blackwell, Bushra Wali, Yijin Xiang, Ali Alawieh, Iqbal Sayeed, Andrew Reisner

**Affiliations:** 1Department of Neuropsychology, Children’s Healthcare of Atlanta, Atlanta, GA 30322, USA; 2Department of Pediatrics, Emory University, Atlanta, GA 30322, USA; yijin.xiang@emory.edu (Y.X.); andrew.reisner@choa.org (A.R.); 3Department of Emergency Medicine, Emory University, Atlanta, GA 30322, USA; bwali@emory.edu; 4Department of Neurosurgery, Emory University, Atlanta, GA 30322, USA; ali.mostafa.alawieh@emory.edu; 5National Institute of Health, Bethesda, MD 20892, USA; iqbal.sayeed@nih.gov; 6Department of Neurosurgery, Children’s Healthcare of Atlanta, Atlanta, GA 30322, USA

**Keywords:** pediatric, biomarker, traumatic brain injury, outcomes

## Abstract

Blood based traumatic brain injury (TBI) biomarkers offer additional diagnostic, therapeutic, and prognostic utility. While adult studies are robust, the pediatric population is less well studied. We sought to determine whether plasma osteopontin (OPN) and S100B alone or in combination predict mortality, head Computed tomography (CT) findings, as well as 6-month functional outcomes after TBI in children. This is a prospective, observational study between March 2017 and June 2021 at a tertiary pediatric hospital. The sample included children with a diagnosed head injury of any severity admitted to the Emergency Department. Control patients sustained trauma-related injuries and no known head trauma. Serial blood samples were collected at admission, as well as at 24, 48, and 72 h. Patient demographics, acute clinical symptoms, head CT, and 6-month follow-up using the Glasgow outcome scale, extended for pediatrics (GOSE-Peds), were also obtained. The cohort included 460 children (ages 0 to 21 years) and reflected the race and sex distribution of the population served. Linear mixed effect models and logistic regressions were utilized to evaluate the trajectory of biomarkers over time and predictors of dichotomous outcomes. Both OPN and S100B correlated with injury severity based on GCS. S100B and OPN showed lower AUC values (0.59) in predicting positive head CT. S100B had the largest AUC (0.87) in predicting mortality, as well as 6-month outcomes (0.85). The combination of the two biomarkers did not add meaningfully to the model. Our findings continue to support the utility of OPN as a marker of injury severity in this population. Our findings also show the importance of S100B in predicting mortality and 6-month functional outcomes. Continued work is needed to examine the influence of age-dependent neurodevelopment on TBI biomarker profiles in children.

## 1. Introduction

Traumatic brain injury (TBI) is a global health problem of epidemic proportions. Worldwide, approximately 70 million individuals sustain a head injury annually [1]. In the US, the highest rates of TBI-related emergency department (ED) visits by age group fall in younger children and adolescents, which are 2–4 fold higher than the incidence in adults [2,3]. The aftermath of TBI can be devastating, including impaired cognition and emotional and behavioral functioning, which increases family burden and negatively impacts quality of life.

The management of TBI is largely predicated on presenting clinical features and imaging phenotypes, typically head CT findings [4]. Despite the implementation of standardized management guidelines in the acute phase, management of sub-acute and chronic phases along with the clinical course and outcomes following injury are notoriously heterogenous [5,6,7]. This unpredictability speaks to the differences in innate biological factors of the patient, as well as the treatment response and variability in recovery trajectories. The situation is further complicated by the lack of a reliable quantitative biomarker of the underlying pathology, as there has been for other diseases such as diabetes and myocardial infarction [8]. Further, the majority of biomarker research has focused on diagnostic blood biomarkers of acute TBI, within the first 24 h after injury, where few candidates have been identified for the diagnosis of subacute or chronic sequelae of TBI [9,10,11,12]. In pediatrics, this variability is further compounded in children given the differences in mechanisms of injury and the age-dependent neurodevelopmental changes and plasticity, particularly during infancy [13,14].

To mitigate these clinical challenges, emerging data has shown that the incorporation of blood-based biomarkers have the potential to refine TBI categorization, direct management and predict outcomes. Currently, there are several candidate blood-based biomarkers that are at varying levels of development, validation, and clinical implementation for acute TBI. The first combination TBI biomarkers to receive FDA approval in acute TBI are glial fibrillary acidic protein (GFAP) and ubiquitin carboxyl terminal hydrolase L1 (UCH-L1) [15,16,17,18,19,20]. These markers have been most reliable in predicting intracranial injury when assessed just after injury. However, more limited markers have shown promise in predicting long-term outcomes. S100B, a marker of astrocyte injury and death, has a high sensitivity for TBI, with higher levels associated with worse TBI severity and poorer outcomes [21]. However, S100B, in addition to other acute and subacute markers of injury, have had less reliability in clinical utility in young children. Given the deleterious effects of radiation in the young, this ability to predict intracranial injury would be especially advantageous in pediatrics [22]. Despite the prevalence of pediatric TBI, studies on the efficacy of biomarkers as predictors of outcome in this population remain limited [9].

In a pilot translational study involving a murine TBI model and a small cohort of pediatric patients, our study team identified osteopontin (OPN), a phosphoprotein chiefly secreted by macrophages and/or activated microglia, as a novel TBI biomarker [23]. In children with TBI, plasma levels of OPN, but not GFAP, correlated with severity of injury and presence of intracranial lesions. Plasma OPN levels continued to increase over the course of 72 h in severe pediatric TBI patients, and predicted mortality, days requiring ventilator, and intensive care unit support.

Given our prior work, we conducted a new prospective investigation to validate the use of OPN as a novel blood-based biomarker in a large cohort of pediatric TBI. Our goal was to expand upon the growing pediatric literature on biomarkers in TBI by utilizing a large single center design and using plasma biomarkers in predicting both acute and subacute outcomes. We specifically examined the prognostic role of OPN and compared it to a well-examined biomarker, S100B, in predicting head CT findings, mortality, and 6-month follow-up functional ratings.

## 2. Materials and Methods

### 2.1. Participants and Study Design

This is a prospective cohort study conducted between March 2017 and June 2021 on pediatric patients presenting with head injury to two of our EDs at a tertiary children’s hospital, one of which is a level I pediatric trauma center. IRB approval was obtained by our institution, which included a waiver of consent to collect the first blood sample just after admission, within 24 h of injury, and prior to legally authorized representative consent due to the traumatic nature of the injury. Informed consent was obtained for all patients within the 24-h window from admission and prior to the second blood draw (Figure 1). If families declined to participate in the study during the consent process, they were asked if the admission sample could still be used; in the majority of the cases, this was approved. Patients were enrolled on a continual basis and included staffing 7 days per week. Only patients with documented moderate and severe TBIs (GCS score 3–12) were followed up at 6 months via phone call. Inclusion criteria involved children and adolescents less than 21 years of age with a documented head injury, as defined within our ED setting by attending physicians, as well as patients entering the ED within 24 h of injury. Children with other forms of physical trauma were included as the control group. Penetrating head injury including gunshot wounds as well as atraumatic brain injury were excluded.

### 2.2. Plasma Sample Collection

Blood samples were collected at admission to the ED within 24 h of injury, as well as three follow-up collections at 24, 48, and 72 h after the time of the initial draw, or until the patient was discharged, whichever was earlier. Blood was prepared and processed according to the TBI Common Data Elements (CDE) Biospecimens and Biomarkers Working Group guidelines in the clinical laboratory at our two hospitals. Samples were centrifuged, aliquoted, and immediately frozen at −80 °C and stored in our hospital laboratory before batch couriering to our biorepository at Emory University. Samples were then batch processed in a research laboratory by the research team (IS). Human plasma OPN and S100B were analyzed in duplicate using commercial ELISA kits (R&D Systems, Minneapolis, MN, USA). According to the manufacturer’s protocol, the human plasma samples were diluted with the dilution buffer provided in the kit. If the ELISA readouts of diluted samples were higher than the assay range, these samples were further diluted until covered by the calibration range.

### 2.3. Clinical Outcomes

Demographic and clinical data were extracted from the hospital electronic health records and double entered for verification. Severity of injury was determined based on the lowest GCS score within 24 h determined by the trauma team and categorized into mild TBI (GCS 13–15), mild–complicated TBI (GCS 13–15 + skull fracture or intracranial injury), moderate TBI (GCS 9–12), and severe TBI (GCS 3–8). Positive head CT was determined based on any acute pathology related to the head injury, including skull fracture, hemorrhage, or other intracranial findings. Functional outcomes were assessed using the seven-item Glasgow Outcome Score, extended pediatric (GOSE-peds), where 8 = upper good recovery (Upper GR), 7 = lower good recovery (Lower GR), 6 = upper moderate disability (Upper MR), 5 = lower moderate disability (Lower MR), 4 = upper severe disability (Upper SD), 3 = lower severe disability (Lower SD), 2 = vegetative state (VS), 1 = death. Favorable outcome was operationalized as a GOSE-peds score of 5–8, while unfavorable outcome was operationalized as a GOSE-peds score of 1–4. Of note, all diseased patients, regardless of timing or enrollment, were coded as 1 on the GOSE-peds. The GOSE-peds was assessed prospectively on the phone with caregivers at 6 months (+/− 1 month) after injury.

### 2.4. Statistical Analyses

Statistical significance was set at *p* < 0.05 and two sided. For descriptive analysis, continuous data were presented as the median (IQR-interquartile) and categorical data were presented as frequency (percentage). Differences in demographic and clinical data between patients in four injury severity groups and one control group were assessed using Kruskal–Wallis tests for continuous data and Chi-squared tests (or Fisher’s exact test when appropriate) for categorical data. Effect sizes were additionally calculated, as standardized mean differences (SMDs), and interpreted using Cohen’s d thresholds, 0.2 (small), 0.5 (moderate), and 0.8 (large). Associations between biomarkers (at admission and at 72-h blood draws) and patients’ demographics were assessed using Kruskal–Wallis tests for categorical covariates and Pearson correlations for continuous covariates.

To understand the trajectory of biomarkers across four time points, we implemented linear mixed effect models for biomarkers as continuous dependent variables, respectively. Patient-specific intercept was added as a random effect to the model. The model contained fixed effects for measurement time points and severity, as well as random intercept for each individual to accommodate repeated measurements. We also assessed the interaction between severity and follow-up time in the longitudinal model to examine whether the trajectory of biomarkers over time differs between severity groups. To further assess the predictive ability of biomarkers on acute outcomes (i.e., mortality and positive head CT) and long-term outcomes (i.e., GOSE-peds at 6 months), we conducted bivariate analysis between each biomarker and outcomes and multivariable analysis between the combination of biomarkers and outcomes using logistics regressions. Odds ratios with corresponding 95% confidence interval (95% CI) were reported. We use area under the curve (AUC) to quantify the discrimination ability for each model.

Finally, in order to provide clinical insight and applicability, we dichotomized biomarkers with significant univariate associations with outcomes in previous models. Specially, we modeled between biomarkers at admission with acute outcomes (i.e., positive head CT and death) and biomarkers at 72 h with 6-month outcome (i.e., GOSE-peds). We used admission biomarker values to inform clinical decision making with regard to head CT orders. Cutpointr package (https://cran.r-project.org/web/packages/cutpointr/vignettes/cutpointr.html, accessed on 15 August 2022) was used to determine optimal cut points with a target to minimize distance to point (0, 1) on receiver operating characteristic (ROC) space (method = minimize_boot_metric, metric = roc01) and bootstrapping the variability of the optimal cut points (*n* = 1000). All analyses were performed using SAS version 9.4 (SAS Institute, Cary, NC, USA) and R statistical software (version 4.2; R Foundation for Statistical Computing, Vienna, Austria).

## 3. Results

### 3.1. Patient Demographics

A total of 592 patients were admitted to Children’s Healthcare of Atlanta at two hospital locations and assessed for eligibility. Of these, 460 met the inclusion criteria for enrollment and 135 were consented for additional blood draws beyond admission, 85 of which were followed up at 6 months (Figure 1). Patients were predominantly male (65.7%) with a median age of 9.69 years (Table 1). All pediatric age groups were represented in this cohort with bimodal frequency distribution that peaks around 2 and 13 years of age (Figure 2). When classified by severity of injury, 61.7% were mild and mild–complicated TBIs, 11.1% were moderate, and 19.6% were severe TBIs, in addition to 7.6% non-TBI trauma controls. Mechanisms of injury for the TBI patients included falls (32.0%), motor vehicle collisions (27.0%), and being struck by/against something (9.6%). Mortality rate was 3.5%, and 13.0% of subjects were transferred to our in a hospital comprehensive inpatient rehabilitation program.

### 3.2. Plasma Concentrations and Demographic Variables

No relationship was found between biomarker levels and sex at admission (OPN, *p* = 0.541; S100B, *p* = 0.665). Age-related differences were found across all blood draws for OPN (Figure 3). Plasma OPN protein expression was inversely correlated with age at admission (Pearson’s r = −0.38, *p* < 0.001), 24 h (Pearson’s r = −0.42, *p* < 0.001), 48 h (Pearson’s r = −0.35, *p* < 0.001), and at 72 h (Pearson’s r = −0.41, *p* < 0.001). S100B also demonstrated age-dependent differences at 24 h (Pearson’s r = −0.27, *p* < 0.01), 48 h (Pearson’s r = −0.29, *p* < 0.001), and at 72 h (Pearson’s r = −0.27, *p* < 0.02) (Figure 4). When racial differences were examined, subjects of African American/Black race had significantly higher levels of admission OPN (*p* = 0.019) and lower levels of S100B (*p* < 0.001).

The median time between injury and plasma sample collection was 3 h, 53 min (range: 0:22 min–19:33 h). The mean values at admission in OPN and S100B correlated with injury severity as classified based on GCS (Figure 5). Across four time points, the trajectories of OPN levels were significantly different between the severity groups, indicated by the significance of interactions between severity and follow-up time (*p*-value for group × time < 0.001; Figure 5). The trajectories of S100B levels over time were not significantly different between the severity groups (*p*-value for group × time = 0.073).

### 3.3. Plasma Concentrations at Admission and Acute Outcomes

Differences in admission levels of OPN and S100B were found in patients who died at any point in time compared to those who survived (Table 2). Odds ratios were computed based on the odds of unfavored outcome (mortality) with a 100 unit increase of OPN and S100B at admission. On multivariable logistic regression adjusted for age, OPN and S100B levels were associated with higher odds of mortality. OPN had the highest predictive aOR (aOR (OPN) =1.4445, aOR (S100B) = 1.04; Figure 6). Univariate ROC analysis showed that S100B had the highest discrimination for predicting mortality (AUC 0.87 [95% CI: 0.8, 0.94]), compared to OPN (AUC 0.79 [95% CI: 0.66, 0.92]) (Figure 7). Multivariable analysis showed that the combination of all biomarkers had a strong AUC value (AUC 0.85 [95% CI: 0.71, 0.98]), but not more than S100B alone (Table 3). Cut-off values were also computed. The cut-off value for OPN in predicting mortality was greater than or equal to 218.63 (12.5% in group with OPN ≥ 218.63 versus 1.4% in group with OPN < 218.63) with an odds ratio of 9.91 ([95% CI: 3.35, 29.36], *p* < 0.001), and a sensitivity of 0.69 and a specificity of 0.82 (Table 2). S100B had a cut-off value of greater than or equal to 1951.35 (16.0% in group with S100B ≥ 1951.35 versus 1.1% in group with S100B < 1951.35) with an odds ratio of 17.57 ([95% CI: 5.49, 56.2], *p* < 0.001) and a sensitivity of 0.75 and a specificity of 0.85.

Admission levels of OPN predicted intracranial findings on the initial head CT (Table 2). Admission biomarker levels were higher in patients with intracranial findings on the head CT compared to those without. Univariate analysis confirmed that OPN had the highest discrimination for predicting the presence of CT abnormalities (AUC 0.57 [95% CI: 0.52, 0.62]), compared to S100B (AUC 0.51 [95% CI: 0.46, 0.56]). Multivariable analysis showed that the combination of all biomarkers increased the AUC value (AUC 0.62 [95% CI: 0.57, 0.68]). Cut-off values were also computed for biomarkers with significant relationships. The cut-off value for OPN at admission in predicting intracranial findings on the head CT was greater than or equal to 144.75 (65.0% versus 53.1%) with an odds ratio of 1.64 ([95% CI: 1.10, 2.44], *p* = 0.015) and a sensitivity of 0.58 and a specificity of 0.54. S100B at admission had a cut-off value of greater than or equal to 492.90 (58.5% versus 59.3%) with an odds ratio of 0.97 ([95% CI: 0.66, 1.43], *p* = 0.875) and with a sensitivity of 0.48 and a specificity of 0.51 (Table 2).

### 3.4. Plasma Concentrations of Biomarkers and GOSE-Peds at 6 Months

OPN and S100B levels at 72 h were higher in patients with unfavorable outcome at 6 months (GOSE-peds < 4) compared to those with favorable outcomes (GOSE-peds > 5), respectively (Table 2). Univariate analysis confirmed that S100B had the highest discrimination for predicting long term outcome (AUC 0.85 [95% CI: 0.72, 0.99]) compared to OPN (AUC 0.76 [95% CI: 0.58, 0.95]). The combination of the two markers did not provide a higher level of predictive power compared to individual biomarkers alone (AUC 0.83 [95% CI: 0.64, 1.0]). Cut-off values were also computed for biomarkers with significant relationships. For OPN at 72 h, the cut-off value for GOSE-peds was greater than or equal to 497.89 (75.0% versus 18.5%) with an odds ratio of 13.2 ([95% CI: 2.59, 67.23], *p* = 0.002), while S100B at 72 h had a cut-off value of greater than or equal to 179.65 (71.4% versus 14.8%), with an odds ratio of 14.37 ([95% CI: 2.98, 69.25], *p* < 0.001), and with a sensitivity of 0.71 and a specificity of 0.85 (Table 2).

## 4. Discussion

This prospective study extends upon our prior work, validating the utility of OPN as a possible blood-based TBI biomarker in a pediatric population. We investigated OPN and compared it to S100B, a well-studied acute phase TBI biomarker that has demonstrated clinical utility in adult populations and some pediatric cohorts. We showed that within 24 h of injury, both biomarker values were significantly elevated compared to trauma controls and were related to injury severity, as measured by GCS. Higher levels of OPN were strongly related to mortality, as well as positive CT findings. Higher levels of S100B and, to a lesser degree OPN, were predictive of a worse 6-month outcome as measured by the GOSE-peds.

In our prior pilot study, we found that OPN was a predictor of acute outcomes after pediatric TBI. This large prospective cohort confirms these findings regarding the utility of OPN in pediatric TBI. Specifically, we found that OPN collected at 72 h post-admission had the most robust relationship with TBI severity, suggesting that it may be better at monitoring the progression of brain injury, as well as the response to interventions. OPN is a small integrin-binding ligand N-linked glycoprotein (SIBLING) that mediates cell-matrix interactions. Immediately following a TBI, there is activation of microglia and macrophages with upregulation of OPN, also called secreted phosphoprotein 1 (SPP1) [24,25]. The attributes of OPN that make it a candidate TBI biomarker include low background plasma levels in healthy conditions, quickly up-regulated in macrophages and/or activated microglia in the injured brain, resulting in a half-life of 13 min, high stability in biofluids, and efficient brain-to-blood transport, presumably due to its integrin-binding property [26]. To date, OPN has not been investigated in adult TBI populations.

The variability in relationships between biomarkers and head CT may be related to several factors. In this study, we deliberately included skull fractures in the definition of positive head CT. In pediatrics, skull fractures are more relevant, as they may be signs of potential abusive head trauma, require further treatment, and have the capacity to develop delayed complications such as growing skull fractures [27]. In a retrospective analysis of TBI registry data from 1206 consecutive non-penetrating TBI patients treated at a Level 1 adult and pediatric trauma center, Sarkar and colleagues noted a higher incidence of skull fractures (44% vs. 20.71%) and epidural hematoma (17.3% vs. 9.8%) among children under 15 years of age compared to adults. The authors also found that children had a statistically lower incidence of contusion, subdural hematoma, subarachnoid hemorrhage, or compression of basal cisterns compared to their adult counterparts [28]. The inclusion of skull fractures rather than just intracranial findings is also discrepant from primary adult studies that typically use this definition [12,29].

This study also found that a multi-marker approach, combining OPN and S100B as predictors, did not increase the diagnostic value for head CT findings or other acute markers compared to biomarkers alone. OPN was related to mortality and findings on head CT, whereas increased levels of S100B was only related to 6-month functional outcomes. These markers both revealed higher odds ratios as individual markers compared to the panel combinations of markers. Still, these observations do not preclude the potential utility of a biomarker panel approach to predict or track response to treatment and recovery over time. There is a clear need for larger scale and multi-institutional studies to investigate biomarker profiles in pediatrics TBI, their prognostic potential, and their differences from adult TBI.

Results revealed strong correlations between biomarker values and age, suggesting that typical age-related variation in these biomarker levels is likely, regardless of the severity of injury. Prior studies have shown that many circulating biomarkers change with age, independent of disease [30,31]. The ability to characterize age-adjusted distributions of these biomarkers in both children who are healthy and following general trauma is critically important to increase specificity of diagnosis using these markers and to be able to inform treatment and management. It is likely that part of the heterogeneity in outcomes following pediatric TBI is due, in part, to the combination of the effects of age and the relationship to the injury and recovery.

Weak correlations were found between plasma biomarkers and 6-month outcomes. This finding may be due, in part, to the use of the GOSE-peds measure which is limited in its assessment of functional outcomes. While the measure itself is considered a gold-standard in TBI global outcome assessment and included in recommendations as a common data element [32], it has been widely criticized for having poor reliability, an inability to detect small but potentially meaningful functional changes in patients, and for having significant ceiling effects [33,34]. The heterogeneity in neurocognitive outcomes following pediatric TBI are likely too complex and widespread to be interpreted by this type of dichotomized outcome measure. Therefore, future research should consider examining biomarkers and their association with more comprehensive neurocognitive domains, including measures of attention and executive functioning, processing efficiency, memory, and motor functioning, all of which can be impacted by varying severities of TBI.

The findings are limited by statistical and clinical considerations. The number of enrollments on admission diminished over the subsequent 24-, 48- and 72-h time points. This attrition was due to a combination of factors including desire of guardians and caretakers to withdraw from the study at 24 h, deaths or discharge before 72 h, and children taken into custodial care which precluded further study. Similar difficulties were found and amplified at our six-month follow-up by change in guardianship, especially in victims of abuse, family relocation, and delayed deaths. Because of this, we were also unable to properly stratify our sample by age and more accurately distinguish biomarker profiles in the youngest age groups. Further, there was limitation on the amount of blood drawn from pediatric patients (1.7 milliliters per kilogram per 24 h) with clinical needs taking priority over research. Our control group included trauma controls, which had observed elevations in both biomarker values at admission. We chose this type of control group based on prior studies utilizing both orthopedic and general trauma control groups. Finally, we were also limited by only following our moderate and severe TBI groups at 6-months and did not follow those patients with milder injuries who may have still experienced symptoms and functional deficits.

## 5. Conclusions

This study demonstrates that blood-based biomarkers S100B and OPN are elevated following TBI in children compared to trauma controls and correlate with injury severity. Using a large prospective cohort, we demonstrated that OPN can serve as a biomarker of a positive head CT, as well as mortality, and would be useful in decision making for obtaining CT imaging in the radiation-vulnerable pediatric population. Additionally, our study showed negative correlations between plasma OPN and S100B concentrations and GOSE-peds scores at 6 months, suggesting the prognostic utility of OPN and S100B at long-term follow-up. As biomarker technology is evolving, ongoing effort will continue to characterize the optimal biomarker panel to be added to the routine clinical diagnostic and prognostic arsenal.

## Figures and Tables

**Figure 1 biomedicines-11-02167-f001:**
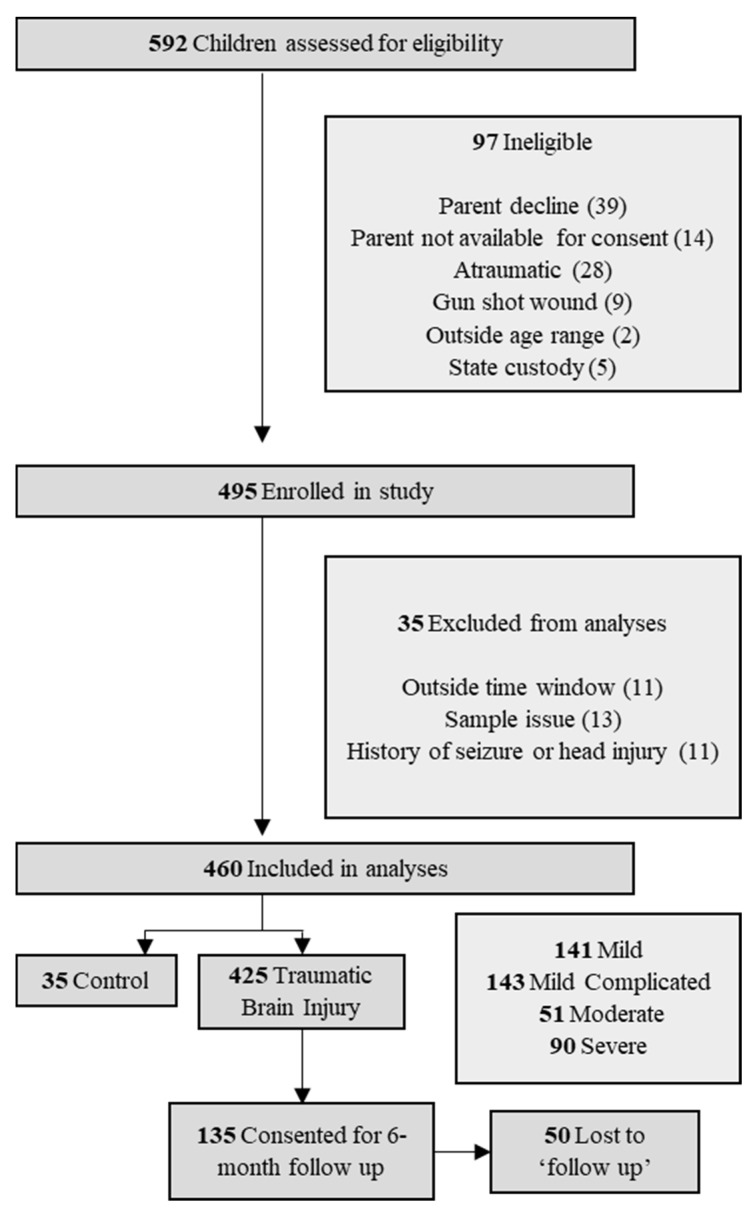
Study flow chart of patient selection.

**Figure 2 biomedicines-11-02167-f002:**
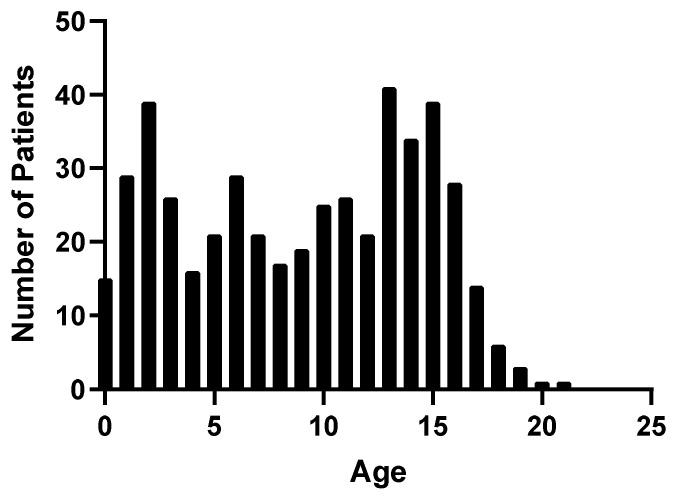
Frequency distribution of TBI patients by age.

**Figure 3 biomedicines-11-02167-f003:**
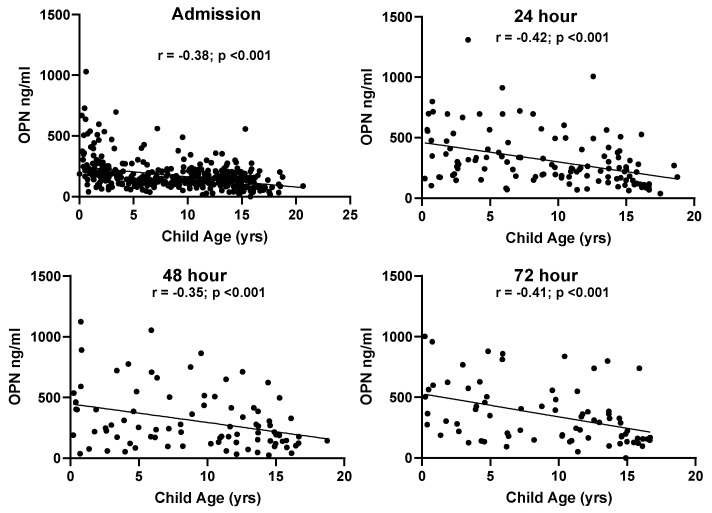
Correlations between patient age and OPN values at all time points in the TBI groups.

**Figure 4 biomedicines-11-02167-f004:**
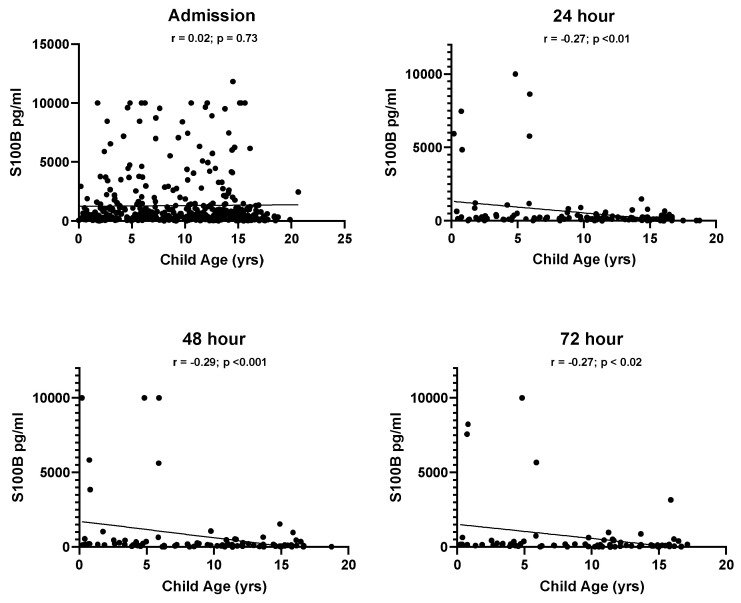
Correlations between patient age and S100B values at all time points in the TBI groups.

**Figure 5 biomedicines-11-02167-f005:**
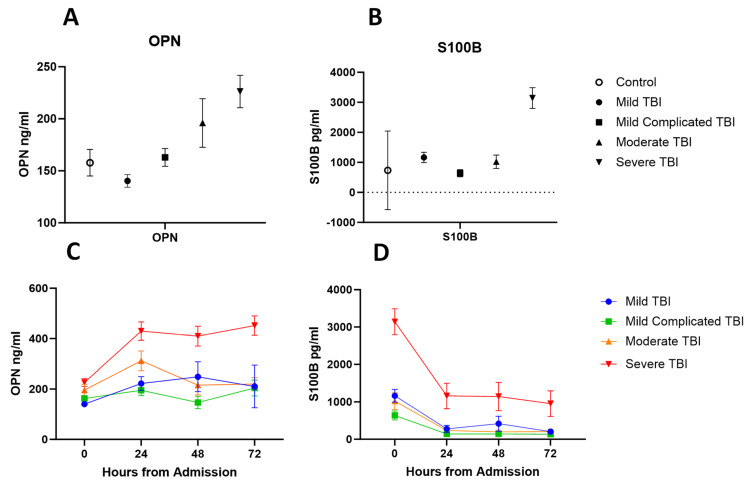
(**A**) Plasma concentration of OPN at admission in trauma controls compared to TBI severity groups. (**B**) Plasma concentration of S100B at admission in trauma controls compared to TBI severity groups. (**C**) Plasma concentration of OPN across 72 h by TBI severity groups. (**D**) Plasma concentration of S100B across 72 h by TBI severity groups. Shown values are for mean and error bars represent SEM.

**Figure 6 biomedicines-11-02167-f006:**
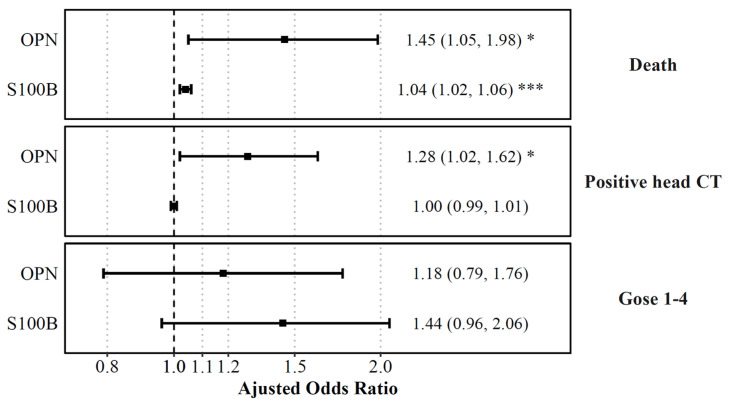
Plasma biomarkers on admission predicting death, positive head CT, and functional outcomes. Displayed are the adjusted odds ratios for each plasma biomarker level on admission, as computed from logistic regression models predicting death, positive head CT or plasma biomarkers levels at 72 h, as computed from a logistic regression model predicting functional outcomes (GOSE) while controlling for age. Shown are the adjusted ORs with error bars representing a 95% confidence interval. The odds ratio indicated the increase of odds of unfavored outcome with a 100 unit increase of OPN, UCH-L1, and S100B, and a 10 unit increase of GFAP. The dashed line is for aOR = 1, and red values represent statistical significance. * *p* < 0.05, *** *p* < 0.001.

**Figure 7 biomedicines-11-02167-f007:**
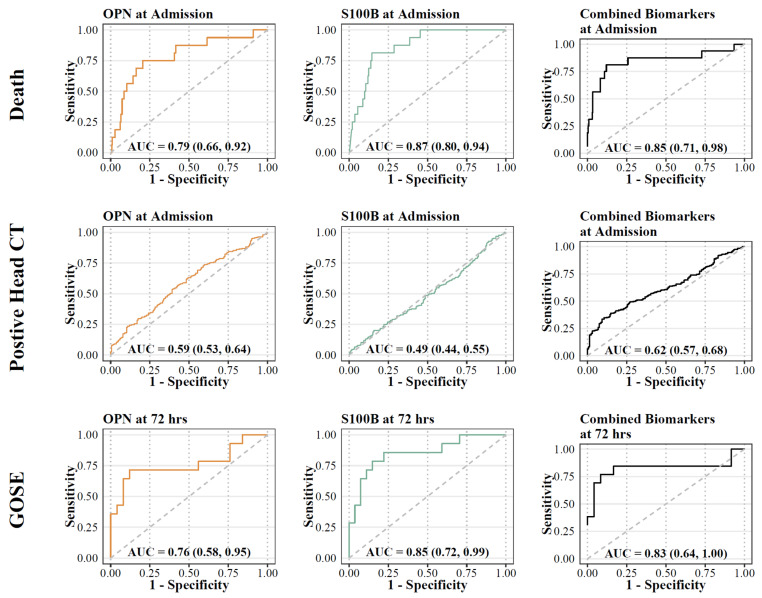
ROC analysis for biomarkers (OPN and S100B) and primary outcomes. Orange lines represent OPN; green lines represent S100B; black lines represent combination.

**Table 1 biomedicines-11-02167-t001:** Admission patient demographic and clinical characteristics ^a^.

	Overall	Control	Mild	Mild-Complicated	Moderate	Severe	*p*	SMD ^b,c^
	*n* = 460	*n* = 35	*n* = 141	*n* = 143	*n* = 51	*n* = 90		
**Patient demographics**								
Age	9.69 [3.92, 13.71]	12.62 [8.20, 14.66]	12.31 [7.58, 14.65]	6.88 [2.05, 13.01]	6.78 [2.25, 13.23]	7.83 [4.07, 11.87]	<0.001	0.43
Sex (M)	302 (65.7%)	24 (68.6%)	89 (63.1%)	96 (67.1%)	36 (70.6%)	57 (63.3%)	0.838	0.086
Ethnicity								
African American or Black	151 (32.8%)	7 (20.0%)	47 (33.3%)	31 (21.7%)	21 (41.2%)	45 (50.0%)	<0.001	0.537
Asian	15 (3.3%)	0 (0.0%)	5 (3.5%)	6 (4.2%)	3 (5.9%)	1 (1.1%)		
Hispanic/ Latino	51 (11.1%)	4 (11.4%)	13 (9.2%)	15 (10.5%)	6 (11.8%)	13 (14.4%)		
Other	9 (2.0%)	0 (0.0%)	1 (0.7%)	3 (2.1%)	3 (5.9%)	2 (2.2%)		
White	234 (50.9%)	24 (68.6%)	75 (53.2%)	88 (61.5%)	18 (35.3%)	29 (32.2%)		
**Clinical variables**								
Mechanism of Injury								
Fall	147 (32.0%)	10 (28.6%)	42 (29.8%)	71 (49.7%)	16 (31.4%)	8 (8.9%)	<0.001	0.967
MVC	124 (27.0%)	10 (28.6%)	43 (30.5%)	16 (11.2%)	13 (25.5%)	42 (46.7%)		
Peds vs. motor vehicle	42 (9.1%)	1 (2.9%)	14 (9.9%)	9 (6.3%)	2 (3.9%)	16 (17.8%)		
Bike	22 (4.8%)	4 (11.4%)	8 (5.7%)	9 (6.3%)	1 (2.0%)	0 (0.0%)		
Struck by/against	44 (9.6%)	2 (5.7%)	15 (10.6%)	18 (12.6%)	6 (11.8%)	3 (3.3%)		
Confirmed abuse	8 (1.7%)	0 (0.0%)	0 (0.0%)	3 (2.1%)	2 (3.9%)	3 (3.3%)		
Suspected abuse	24 (5.2%)	0 (0.0%)	1 (0.7%)	8 (5.6%)	6 (11.8%)	9 (10.0%)		
ATV	32 (7.0%)	1 (2.9%)	15 (10.6%)	8 (5.6%)	4 (7.8%)	4 (4.4%)		
Other	17 (3.7%)	7 (20.0%)	3 (2.1%)	1 (0.7%)	1 (2.0%)	5 (5.6%)		
**Clinical outcomes**								
Death	16 (3.5%)	0 (0.0%)	0 (0.0%)	0 (0.0%)	0 (0.0%)	16 (17.8%)	<0.001	0.263
Positive head CT	252 (58.9%)	0 (0.0%)	11 (9.4%)	132 (92.3%)	34 (68.0%)	75 (83.3%)	<0.001	1.854
ICU Admission	184 (40.0%)	2 (5.7%)	10 (7.1%)	46 (32.2%)	38 (74.5%)	88 (97.8%)	<0.001	1.791

^a^ Standardized mean differences (SMD) is the absolute difference in means or proportions divided by standard error; imbalance is defined as absolute value greater than 0.20 (small effect size); effect size indices of 0.2, 0.5, and 0.8 can be used to represent small, medium, and large effect sizes. ^b^ ANOVA tests were employed for parametric analysis. ^c^ Kruskal–Wallis tests were employed for non-parametric analysis. Abbreviations: MVC, motor vehicle collision; Peds, Pediatrics; ATV, all-terrain vehicle; ICU, intensive care unit.

**Table 2 biomedicines-11-02167-t002:** Descriptive analyses for primary outcomes and predictor biomarkers.

	≥Cut-Off	<Cut-Off	Odds Ratio (95% CI)	AUC (95% CI)	Sensitivity (95% CI)	Specificity (95% CI)
**OPN at admission**						
**Death**	≥218.63	<218.63				
Yes	11 (12.5%)	5 (1.4%)	9.91 (3.35, 29.36)	0.75 (0.63, 0.87)	0.69 (0.46, 0.91)	0.82 (0.78, 0.86)
No	77 (87.5%)	347 (98.6%)	Ref.			
**Positive head CT**	≥144.75	<144.75				
Yes	141 (65.0%)	102 (53.1%)	1.64 (1.10, 2.44)	0.56 (0.51, 0.61)	0.58 (0.52, 0.64)	0.54 (0.47, 0.62)
No	76 (35.0%)	90 (46.9%)	Ref.			
**OPN at 72 h**						
**Gose**	≥497.89	<497.89				
1–4	9 (75.0%)	5 (18.5%)	13.2 (2.59, 67.23)	0.76 (0.62, 0.91)	0.64 (0.39, 0.89)	0.88 (0.75, 1)
5–8	3 (25.0%)	22 (81.5%)	Ref.			
**S100B at admission**						
**Death**	≥1951.35	<1951.35				
Yes	12 (16.0%)	4 (1.1%)	17.57 (5.49, 56.2)	0.8 (0.69, 0.91)	0.75 (0.54, 0.96)	0.85 (0.82, 0.89)
No	63 (84%)	369 (98.9%)	Ref.			
**Positive head CT**	≥492.90	<492.90				
Yes	117 (58.5%)	128 (59.3%)	0.97 (0.66, 1.43)	0.5 (0.45, 0.55)	0.48 (0.42, 0.54)	0.51 (0.44, 0.59)
No	83 (41.5%)	88 (40.7%)	Ref.			
**S100B at 72 h**						
**Gose**	≥179.65	<179.65				
1–4	10 (71.4%)	4 (14.8%)	14.37 (2.98, 69.25)	0.78 (0.64, 0.92)	0.71 (0.48, 0.95)	0.85 (0.72, 0.99)
5–8	4 (28.6%)	23 (85.2%)	Ref.			

Note: The threshold adopted to binarize the level of biomarkers was selected using a bootstrapping approach (*n* = 1000 replicates) to select the cut point minimizing the distance of the ROC curve to the point of perfect discrimination (0, 1) with the cutpointr R package.

**Table 3 biomedicines-11-02167-t003:** Univariate and multivariate regression models between individual and combined biomarkers and outcomes. Admission levels were used for death and head CT variables; 72 h levels were used for the GOSE-peds variable.

	OPN	S100B
**Death**		
Univariable Analysis		
OR (95% CI)	1.65 (1.27, 2.16) ***	1.03 (1.02, 1.05) ***
AUC	0.79 (0.66, 0.92)	0.87 (0.80, 0.94)
Multivariable Analysis		
OR (95% CI)	1.45 (1.05, 1.98) *	1.04 (1.02, 1.06) ***
AUC	0.85 (0.71, 0.98)	
**Head CT**		
Univariable Analysis		
OR (95% CI)	1.44 (1.16, 1.78) ***	1 (0.99, 1.01)
AUC	0.59 (0.53, 0.64)	0.49 (0.44, 0.55)
Multivariable Analysis		
OR (95% CI)	1.28 (1.02, 1.62) *	1.00 (0.99, 1.01)
AUC	0.62 (0.57, 0.68)	
**GOSE-Peds**		
Univariable Analysis		
OR (95% CI)	1.56 (1.15, 2.11) **	1.46 (1.01, 2.11) *
AUC	0.76 (0.58, 0.95)	0.85 (0.72, 0.99)
Multivariable Analysis		
OR (95% CI)	1.18 (0.79, 1.76)	1.44 (0.96, 2.06)
AUC	0.83 (0.64, 1.00)	

Note: Notation for significance: * *p*-value < 0.05; ** *p*-value < 0.01; *** *p*-value < 0.001. Within the strata of measurement time, the largest AUC was highlighted. The odds ratio indicated the increase of odds of an unfavored outcome with a 100 unit increase of OPN and S100B.

## Data Availability

De-identified data can be made available through contacting the corresponding author and through appropriate data use agreements determined by institutions.

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
