# Peer review of "Prognostic Value of Plasma Biomarkers S100B and Osteopontin in Pediatric TBI: A Prospective Analysis Evaluating Acute and 6-Month Outcomes after Mild to Severe TBI"

_biomedicines, 2023, doi:10.3390/biomedicines11082167_

Round 1

Reviewer 1 Report

Thank you for nominating me as a reviewer. In this paper, authors sought to determine whether plasma Osteopontin (OPN) and S100B alone or in combination predict mortality, head CT findings, as well as 6-month functional outcomes after TBI in children. This is a prospective, observational study between March 2017 and June 2021 at a tertiary pediatric hospital. The sample included children with a diagnosed head injury of any severity admitted to the Emergency Department. Control patients sustained trauma related injuries and no known head trauma. Serial blood samples were collected at admission, 24, 48, and 72 hours. The quality of the study would be better if the authors completed minor revisions.

1, The Introduction section is well written, but some paragraphs could be moved to the Methods section or better explained by the authors. For example, lines 69-76 could be moved to the Methods section or better explained by the authors.

2, Figure 1 is well presented, but the number of excluded subjects does not match the description. For example, 35 subjects were excluded from the study's process, but the total number of excluded subjects is 36.

3, line 174-184: If possible, the authors should be more specific in describing their statistical testing methods.

4. Is Figure 2 essential to this study?

5. The authors should add limitations in the discussion section.

Author Response

Please see attached cover letter. 

Reviewer 2 Report

Authors investigated the prognostic value of two markers S100B and Osteopontin in plasma in pediatric TBI. Serial blood samples were collected at admission, 24, 48, and 72 hours after TBI in children. They found that both OPN and S100B correlated with injury severity based on GCS. S100B had the largest AUC (0.87) in predicting mortality as well as 6-month outcomes (0.85). The combination 25 of the two biomarkers did not add meaningfully to the model. They concluded that S100B has an important role in predicting mortality and 6-month functional outcomes. It is an interesting study but there are important issues that are needed to be considered:

1.     They included cases from 0-21 years old, however, cases older than 16 years old are not considered as children, therefore, authors need to exclude these cases and run analysis , otherwise it will not be considered for pediatric TBI.

2.     Results show significant negative correlation between the age and plasma levels of both S100B and Osteopontin, therefore, it is important to include cases in the age of younger than 16 years.

3.     It is not clear why authors selected levels of biomarkers only from 72 hours after admission to correlated with CT findings?

4.     It is clear that most of the cases were male, therefore it is important to consider sex and perform analysis to see if there is a sex difference in the prognostic values of theses 2 markers.

5.     From figure 5, it is clear that the level of S100B in mild and moderate TBI is lower than the control-non head trauma. How authors explain it? How they could consider it as the biomarker for TBI?

Author Response

Please see cover letter attached.

Round 2

Reviewer 2 Report

Authors responded to the comments in a satisfactory manner.